# A Framework to Assess the Information Dynamics of Source EEG Activity and Its Application to Epileptic Brain Networks

**DOI:** 10.3390/brainsci10090657

**Published:** 2020-09-22

**Authors:** Ivan Kotiuchyi, Riccardo Pernice, Anton Popov, Luca Faes, Volodymyr Kharytonov

**Affiliations:** 1Department of Biomedical Engineering, National Technical University of Ukraine “Igor Sikorsky Kyiv Polytechnic Institute”, 03056 Kyiv, Ukraine; ivanellokot@gmail.com; 2Data & Analytics, Ciklum, London WC1 A 2TH, UK; popov.kpi@gmail.com; 3Department of Engineering, University of Palermo, 90133 Palermo, Italy; riccardo.pernice@unipa.it; 4Department of Electronic Engineering, National Technical University of Ukraine “Igor Sikorsky Kyiv Polytechnic Institute”, 03056 Kyiv, Ukraine; 5Clinical Hospital “Psychiatry”, 03056 Kyiv, Ukraine; vkharytonov69@gmail.com

**Keywords:** epilepsy, information theory, EEG, information storage, information transfer, vector autoregressive modeling, common spatial patterns, independent component analysis

## Abstract

This study introduces a framework for the information-theoretic analysis of brain functional connectivity performed at the level of electroencephalogram (EEG) sources. The framework combines the use of common spatial patterns to select the EEG components which maximize the variance between two experimental conditions, simultaneous implementation of vector autoregressive modeling (VAR) with independent component analysis to describe the joint source dynamics and their projection to the scalp, and computation of information dynamics measures (information storage, information transfer, statistically significant network links) from the source VAR parameters. The proposed framework was tested on simulated EEGs obtained mixing source signals generated under different coupling conditions, showing its ability to retrieve source information dynamics from the scalp signals. Then, it was applied to investigate scalp and source brain connectivity in a group of children manifesting episodes of focal and generalized epilepsy; the analysis was performed on EEG signals lasting 5 s, collected in two consecutive windows preceding and one window following each ictal episode. Our results show that generalized seizures are associated with a significant decrease from pre-ictal to post-ictal periods of the information stored in the signals and of the information transferred among them, reflecting reduced self-predictability and causal connectivity at the level of both scalp and source brain dynamics. On the contrary, in the case of focal seizures the scalp EEG activity was not discriminated across conditions by any information measure, while source analysis revealed a tendency of the measures of information transfer to increase just before seizures and to decrease just after seizures. These results suggest that focal epileptic seizures are associated with a reorganization of the topology of EEG brain networks which is only visible analyzing connectivity among the brain sources. Our findings emphasize the importance of EEG modeling approaches able to deal with the adverse effects of volume conduction on brain connectivity analysis, and their potential relevance to the development of strategies for prediction and clinical treatment of epilepsy.

## 1. Introduction

The human brain is an extremely large-scale network of neurons, whose function depends on mutual interactions continuously occurring between different sources of neuronal activity. Characterization of the brain function is often conducted through the analysis of functional connectivity, which reflects temporal correlations between the brain dynamics of spatially separated regions, and effective connectivity, which describes networks of directional effects of one neural element over another [1]. This analysis is applied in particular through processing of electroencephalogram (EEG) signals, which record voltage differences between various locations on the head surface, through measuring the strength of interactions between close or remote brain areas. Brain connectivity analysis is often performed to investigate functional alterations of the brain activity in a wide variety of neurological disorders [2,3,4]. In this context, the evaluation of connectivity patterns is of particular relevance to disorders of the central nervous system related to epilepsy, as seizure attacks typically produce redundant hyper-synchronous activity of neurons in the brain [5]. In fact, estimating connectivity in the epileptic brain may lead to a better understanding of the occurrence and spreading of epileptiform activity, and is of relevance for different research and clinical applications [3,6,7]. The importance of such approach is even more strengthened by recent increased evidence of seizure onset not in the entire brain (generalized seizures) or in a circumscribed region of the brain (focal seizures), but within a network of brain regions (the so-called “epileptic network” composed of several functionally connected cortical and subcortical brain structures and regions) [8].

The assessment of brain connectivity is a challenging issue which has been addressed in the literature using a wide range of signal processing techniques including time domain methods, time/frequency approaches, parametric models, phase synchronization and nonlinear measures [9,10,11]. Among them, the use of vector autoregressive (VAR) models is very popular, thanks to the simplicity of the linear parametric representation, to its flexibility (no assumption is made about the underlying neural mechanisms), and to the possibility to exploit it in the assessment of coupling and directed interactions in time, frequency and information-theoretic domains [12,13]. In particular, the connection of the VAR representation to the ubiquitous concept of Granger causality (GC) and to extensions of this concept developed in the framework of information dynamics [14,15] makes this approach eligible for the causal analysis of functional brain connectivity in several applicative contexts including the characterization of epileptic activity [3]. It is worth remarking that ‘causal’ interpretations of Granger causality measures must be done cautiously, as these measures reflect a statistical and purely data-driven concept of causality. In particular, GC measures are designed to reflect the causal effect that the underlying physical mechanisms have on the analyzed neuroimaging data, and neither functional nor effective connectivity representations necessarily map univocally onto the underlying anatomical (structural) connectivity [16].

The simplest and most intuitive way to study brain connectivity is to apply connectivity measures directly on the signals recorded from the scalp EEG sensors. However, scalp-level brain connectivity analysis suffers from several problems, including the fact that the position of scalp EEG electrodes do not relate trivially to the location of the underlying sources [17], thus becoming mostly sensitive to activity correlated over large areas of the superficial cortical surface, with smaller contributions from deeper sources, the effects on the scalp recordings of the so-called normal EEG background consisting in a common electrical activity shared together with presumably different activity in comparable EEG classes, and the presence of recording artifacts which are generally assumed to be components of extracerebral origin [18,19]. Besides that, the scalp EEG signals are influenced by the well-known problem of volume conduction, which refers to the mixing effects which result from measuring electrical potentials at a distance from their source generators [20]. Due to volume conduction, signals originating from the same source in the brain are detected by several EEG electrodes. As a result, VAR-based connectivity measures applied straightly to scalp EEG signals can lead to the detection of spurious causality and more generally of wrong connectivity patterns [21,22].

A solution to limit the adverse effects of volume conduction on EEG connectivity analysis is to transfer the connectivity estimation problem from the scalp sensor domain to the cortical source domain [23]. This is typically achieved first reconstructing the source electrical activity and then applying the desired connectivity metrics on the estimated source time series. Although it is acknowledged that this approach is useful to limit the issue of volume conduction, it has also been criticized or even defined “elusive” in the past due to the lack of uniqueness of the reconstructed sources (e.g., depending on the method applied to reconstruct the sources, or the number of sources employed, or even on the portion of the time series used) since the relationship between each notion of functional connectivity and its underlying neural substrate is still unknown [24]. Source reconstruction can be based on source localization methods, which allow spatial analysis but require accurate models of anatomy and electrical properties of the head [25], or by blind decomposition methods such as Principal Component Analysis (PCA) or Independent Component Analysis (ICA), which do not require head modeling and yield source signals which can be interpreted as originating from cortical dipoles [26]. A more accurate approach, which combines the steps of source signal estimation and connectivity analysis, is that proposed in [27] who integrated ICA source reconstruction and VAR connectivity analysis. This approach, known as VARICA source connectivity analysis, has been further refined through its integration with the Common Spatial Patterns (CSP) technique for dimensionality reduction, leading to the CSPVARICA method [28]. The use of CSP allows to retain only the EEG components which explain the difference in variance between two conditions, and it is thus particularly useful when differences between states (e.g., baseline/task or pre/post) are analyzed. The present work introduces a modified implementation of the SCoT toolbox for CSPVARICA EEG source connectivity analysis [28] and reports its evaluation on simulations of interacting VAR cortical source processes mixed to obtain synthetic EEG signals, as well as its application to the characterization of brain connectivity in children manifesting episodes of generalized or focal seizures. The new implementation allows, besides the reconstruction of source EEG activities from scalp recordings based on improved CSP and ICA algorithms, the computation of the measures of information dynamics (information storage and information transfer with statistical significance assessment) performed in both sensor and source domains. Moreover, the comparison of scalp and cortical source information dynamics computed in epileptic children leads us to make inferences about the reorganization of the brain connectivity networks that occurs before and after epileptic seizures, distinguishing between generalized or focal seizures and addressing the role of volume conduction.

The algorithms of the framework for the quantification of EEG source information dynamics and evaluation on simulations presented in this work are collected and can be reproduced within the CSPVARICA Information Dynamics MATLAB toolbox, which is available as supplementary material of this article and can be freely downloaded from www.lucafaes.net/CSPVARICAInfoDyn.html.

## 2. Framework for the Quantification of EEG Source Information Dynamics

The methodology proposed in our work for assessing information dynamics is illustrated in Figure 1. Considering two experimental conditions (classes $H=\left\{h_{1},h_{2}\right\}$) and a given number of trials per condition ($K=\left\{K_{1},K_{2}\right\}$), the starting point is a dataset of EEG time series of length *N* acquired from *D* scalp sensors for each trial and condition. The procedure consists of the following steps: (i) application of CSP [29] to select, across all trials, the *Q* reduced EEG components which better explain the difference in variance between the two conditions; (ii) VAR modeling [13] of the reduced components, performed at a single trial level, to describe the time-lagged linear interactions among the EEG components; instantaneous interactions not explained by lagged effects are retained in the VAR residuals; (iii) application of ICA [26] to the VAR residuals, concatenated over trials and classes to get an overall description of instantaneous EEG influences related to volume conduction; ICA returns the unmixing matrix that relates the instantaneously independent cortical sources to the reduced EEG components; (iv) unmixing of the EEG time series to get the source time series for each single trial; this step exploits the overall unmixing matrix relating the source and scalp dynamics, which is obtained combining the ICA unmixing matrix and the CSP filter matrix [28]; (v) VAR modeling of the source time series, performed at a single trial level, to describe the time-lagged linear interactions among the instantaneously independent EEG sources [27]; (vi) inverse solution of the Yule-Walker Equations for the VAR source parameters, to reconstruct the autocorrelation sequence of the source time series [14]; (vii) definition of sub-models for each predefined source time series, to describe its predictability from its own dynamics or from the dynamics of the other sources in terms of partial variances [30], and subsequent computation of the information dynamics measures.

### 2.1. Theoretical Formulation

In a stochastic signal processing setting, a set of EEG signals recorded with sampling frequency $f_{s}$ from *D* locations in the scalp are represented as a realization of a *D*-dimensional discrete time random process, $x_{n}={\left[x_{1,n}\ldots x_{D,n}\right]}^{T},n\in Z$, where the time index *n* identifies the sample acquired at time $n/f_{s}$. To describe volume conduction, the process is modeled as an instantaneous mixing of *Q* cortical sources $s_{n}={\left[s_{1,n}\ldots s_{Q,n}\right]}^{T}$ according to the linear equation:(1)$$x_{n}=Ms_{n}$$
where *M* is a mixing matrix of dimension (D×Q). In turn, the sources are modeled as a VAR model of the following form:(2)$$s_{n}=\sum_{k=1}^{p}B_{k}s_{n-k}+e_{n},$$
where $B_{k}$ are (Q×Q) coefficient matrices describing the source interactions at lag *k*, *p* is the VAR model order, and $e_{n}$ is a *Q*-dimensional vector of white and independent non-Gaussian noise processes with (Q×Q) diagonal covariance matrix $\Sigma_{e}=E\left[e_{n}e_{n}^{T}\right]$.

According to the CSPVARICA method proposed in [28], the redundancy of the scalp recordings can be reduced applying the CSP technique to a set of EEGs collected in two conditions over multiple trials. Specifically, CSP results in *D* spatial filters stored as the rows of a square transformation matrix that relates the original EEG scalp signals $x_{n}$ to the reduced components $y_{n}$, and explains the differences in variance between the two conditions. A subset of *Q* spatial filters is then selected, according to the procedure detailed in Section 2.2, to form the (Q×D) filter matrix *C*. Filtering the *D* original EEG signals $x_{n}$ with such matrix yields the *Q* reduced components $y_{n}$ which retain the largest part of the EEG variance between the two analyzed conditions [31]. The reduced components can be easily related to the source signals $s_{n}$ using the (Q×Q) mixing matrix W=CM [28]:(3)$$y_{n}=Cx_{n}=CMs_{n}=Ws_{n}.$$

Then, given that $s_{n}=W^{-1}y_{n}$ and applying the source model (Equation (Equation 2)), it can be easily shown that the following VAR representation holds for the reduced components:(4)$$y_{n}=\sum_{k=1}^{p}A_{k}y_{n-k}+r_{n},$$
with coefficient matrices $A_{k}=WB_{k}W^{-1}$, and residuals $r_{n}=We_{n}$. Note that, if *W* is known or estimated, the source processes can be obtained from the knowledge of the transformation matrix *C* as follows:(5)$$s_{n}=W^{-1}Cx_{n}={\left(CM\right)}^{-1}Cx_{n}=Ux_{n}.$$

Thus, the overall (Q×D) unmixing matrix that allows the reconstruction of the sources starting from the scalp EEG signals has the form:(6)$$U={\left(CM\right)}^{-1}C.$$

When both the source time series $s_{n}$ and the VAR parameters ($B_{k}$, $\Sigma_{e}$) are known, they can be used to estimate the connectivity measures in the framework of information dynamics. This can be done according to the approach proposed in [32,33], which first determines the autocovariance sequence of the source process (Equation (Equation 2)), then arranges the elements of the autocovariance matrix to compute the so-called “partial variances” which reflect the unpredictability of one source given its own dynamics or the dynamics of the other sources, and finally computes network information measures exploiting the relation between (partial) variance and (conditional) entropy. Specifically, the first step is accomplished in solving the inverse Yule-Walker equations [14] for the process (Equation (Equation 2)):(7)$$\Gamma_{k}=\sum_{l=1}^{p}B_{l}\Gamma_{k-l}+\delta_{k0}\Sigma_{e},$$
where $\Gamma_{k}=E\left[s_{n}s_{n-k}^{T}\right]$ is the (Q×Q) autocovariance matrix of the sources evaluated at lag *k*, and $\delta_{k0}$ is the Kronecker product. The second step is accomplished implicitly forming a VAR submodel having the $j^{th}$ source $s_{j,n}$ as the target and predicting it as a linear combination of its *q* past values $s_{j,n}^{q}=\left[s_{j,n-1},\ldots ,s_{j,n-q}\right],$ and possibly of the *q* past values of one or more other sources. All these past values are included in the regression vector *V*, and then the partial variance of $s_{j,n}$ given *V*, quantifying the variance of the prediction error of the linear regression of of $s_{j,n}$ on *V*, is obtained as [32]:(8)$$\Sigma_{j|V}=\Sigma_{j}-\Sigma_{j,V}\Sigma_{V}^{-1}\Sigma_{j,V}^{T}$$
where $\Sigma_{j}$ is the variance of $s_{j}$, $\Sigma_{j}=E\left[s_{j,n}^{2}\right]$, $\Sigma_{j,V}$ is the covariance between $s_{j,n}$ and *V*, $\Sigma_{j,V}=E\left[s_{j,n}V\right]$, and $\Sigma_{V}$ is the covariance of *V*, $\Sigma_{V}=E\left[VV^{T}\right]$. After separating the *Q* sources $s=[s_{1},\ldots ,s_{Q}]$ to evidence the target $s_{j}$, the driver $s_{i}$ and the other sources $s_{k}\left(k=\left\{1,\ldots ,Q\right\}\backslash \left\{i,j\right\}\right)$, the formulation of Equation (Equation 8) is exploited to compute the partial variances of $s_{j}$ given its own past $\left(\Sigma_{j|j}\right)$ when $V=s_{j,n}^{q}$, given its past and the past of $s_{k}$$\left(\Sigma_{j|j,k}\right)$ when $V=[s_{j,n}^{q}s_{k,n}^{q}]$, and given the past of all *Q* sources $\left(\Sigma_{j|i,j,k}\right)$ when $V=[s_{i,n}^{q}s_{j,n}^{q}s_{k,n}^{q}]$. Finally, these partial variances are used to compute information-theoretic measures of information storage, information transfer and conditional information transfer as [32]:(9)$$\begin{array}{c} S_{j}=\frac{1}{2}\ln\frac{\Sigma_{j}}{\Sigma_{j|j}}\\ T_{j}=\frac{1}{2}\ln\frac{\Sigma_{j|j}}{\Sigma_{j|i,j,k}}\\ T_{i\to j|k}=\frac{1}{2}\ln\frac{\Sigma_{j|j,k}}{\Sigma_{j|i,j,k}} \end{array}$$

The information storage $S_{j}$ (Equation (Equation 9)) can be considered as a measure of self-predictability for the *j*th source, quantifying the amount of information shared between the present and the past dynamics of the source process. The information transfer $T_{j}$ (Equation (Equation 9)), or total Granger causality, is a measure of the amount of information contained in the present state of the *j*th source that can be predicted by the past states of all the other sources. Finally, the conditional information transfer $T_{i\to j|k}$ (Equation (Equation 9)), or conditional Granger causality, is a measure of the amount of information contained in the present state of the *j*th source that can be predicted by the past states of the *i*th source above and beyond the information that can predicted from the other sources. More details about information dynamics measures can be found in [33] and in references therein.

### 2.2. Practical Estimation

The starting point of our analysis is a set of EEG signals collected in two different classes $H=(\left\{h_{1},h_{2}\right\})$ with presumably different brain activity over multiple sets (or trials); for each class and trial, *D* EEG signals are observed simultaneously recording *N* samples per signal, so that a period of duration $T=N/f_{s}$ secs is covered. The analysis steps to estimate the measures of information dynamics at the level of the cortical sources are described in the following.

#### 2.2.1. Common Spatial Patterns (CSP)

The first step is the application of CSP [29,31] in order to find, from the original EEG data matrix of dimension D×N×K×H, the reduced data matrix of dimension Q×N×K×H, with Q<D, which explains the largest part of the difference in variance between the two classes $h_{1}$ and $h_{2}$, as well as the spatial filter matrix *C* that relates these two data matrices (Equation (Equation 3)). Denoting the number of trials belonging to the two classes as $K_{1}$ and $K_{2}$, the CSP algorithm is applied to all the trials of the same class at once.

From the D×N data matrix $x_{k}^{h_{i}}$ arranged to contain as rows the EEG signals acquired for the trial *k* of the class $h_{i}$ ($k=1,\ldots ,K_{i};i=1,2$), CSP estimates the D×D covariance matrices:(10)$$P_{h_{i}}=\frac{1}{K_{i}}\sum_{k=1}^{K_{i}}\frac{x_{k}^{h_{i}}{\left(x_{k}^{h_{i}}\right)}^{T}}{tr(x_{k}^{h_{i}}{\left(x_{k}^{h_{i}}\right)}^{T})}$$
where tr(·) denotes the sum of the diagonal elements of the covariance matrix. Then, a spatial filtering matrix $C_{D}$ is defined having the spatial filters $c_{j}={\left[c_{j,1}\cdots c_{j,D}\right]}^{T}$ as columns (j=1,…,D). This matrix is obtained using a joint diagonalization of the class-related covariance matrices $P_{h_{1}}$ and $P_{h_{2}}$, achieved solving the following generalized eigenvalue problem [31]:(11)$$P_{h_{1}}c_{j}=\lambda_{j}\left(P_{h_{1}}+P_{h_{2}}\right)c_{j}$$
where $\lambda_{j}$ is the eigenvalue associated to each spatial filter $c_{j}$, and is related to the discriminative power of the filter: large eigenvalues correspond to spatial filters providing high variance in one class but low variance in the other, and vice versa. Thus, the spatial filters help to discriminate between the two classes.

When used without any dimensionality reduction, the CSP algorithm exploits all the *D* spatial filters contained in the matrix $C_{D}$. However, it is usual to reduce dimensionality selecting a subset containing the Q<D most relevant filters; the pruned spatial filtering matrix *C*, applied to the EEG data matrices, leads to the reduced components that contribute most to the total EEG variance (see Figure 1 and Equation (Equation 3)). In this work, $C_{D}$ is pruned manipulating the spatial covariance matrices $P_{h_{i}}$ used by the CSP algorithm in a rigorous way exploiting Riemannian geometry [31]. This is achieved analyzing the contribution of the fractional distances $\delta_{j}$ (j=1,…,D) to the total Riemannian distance between the two symmetric positive-definite covariance matrices $P_{h_{1}}$ and $P_{h_{2}}$:(12)$$\delta_{j}\left(P_{h_{1}},P_{h_{2}}\right)=\frac{\log^{2}\left(\frac{\lambda_{j}}{1-\lambda_{j}}\right)}{\sqrt{\sum_{j=1}^{D}\log^{2}\left(\frac{\lambda_{j}}{1-\lambda_{j}}\right)}}$$

According to [31], the fractional distance $\delta_{j}$ and the percentage of total variance described by the spatial filter $c_{j}$ directly depend on the corresponding eigenvalue $\lambda_{j}$. Consequently, to choose the most appropriate number *Q* of filters to discriminate the two classes, a viable criterion is to sum up the fractional distances in order to cover a predefined percentage of the total variance (greater than 90% in this study). After finding *Q* and consequently the transformation matrix *C*, we apply it to the EEG data x in order to reduce the dimensionality of the CSP-filtered signals y (Figure 1, step (i)), and limit the number of sources.

#### 2.2.2. VAR Modeling of Reduced EEG Components

The second analysis step consists in describing the interactions among the reduced EEG components in terms of VAR modeling. VAR identification is performed on each (Q×N) data matrix $y_{k}^{h_{i}}=Cx_{k}^{h_{i}}$ obtained for any trial *k* of each class $h_{i}$ ($k=1,\ldots ,K_{i};i=1,2$). VAR identification allows to find estimates of the coefficients $A_{k}$ and residuals $r_{n}$ in Equation (Equation 4), collected respectively in the (Q×pQ) matrix $A=[A_{1}\cdots A_{p}]$ and in the (Q×(N−p)) matrix $R=[r_{p+1}\cdots r_{N}]$, after writing Equation (Equation 4) in compact representation as:(13)Y=AZ+R,
where $Y=[y_{p+1}\cdots y_{N}]$ and $Z={\left[Z_{1}^{T}\cdots Z_{p}^{T}\right]}^{T}$ (with $Z_{i}=y_{p-i+1}\cdots y_{N-i}$) are properly arranged observation matrices. Equation (Equation 13) is typically solved through the multivariate least squares method [13], finding the estimates $A=YZ^{T}{\left(ZZ^{T}\right)}^{-1}$ and R=Y−AZ.

VAR identification is completed by selection of the model order and validation of the model assumption of whiteness of the residuals. In this work, the VAR model order is selected as the value of *p* minimizing the cost function implemented by the Schwartz Bayesian Criterion (SBC) [34]. Then, whiteness of the estimated residuals was tested using the multivariate Li-McLeod portmanteau test statistic Q [35] which, under the null hypothesis of white residuals, returns a *p*-value higher than 0.05 in case of whiteness [28,34].

#### 2.2.3. Independent Component Analysis (ICA)

The VAR analysis performed on the CSP-filtered signals $y_{n}$ explains the time-delayed cross-dependencies between these reduced EEG components but cannot describe their instantaneous (zero-lag) dependencies, which are thus retained in the residuals $r_{n}$. Such zero-lag dependencies are typically ascribed to volume conduction [28], in a way such that the reduced EEG components result from the instantaneous mixing of the source signals and, following the VAR models (Equations (Equation 2) and (Equation 4)), the VAR residuals are subjected to the same mixing (i.e., $y_{n}=Ws_{n}$ and $r_{n}=We_{n}$). Therefore, under the assumption that the residuals $e_{n}$ are non-Gaussian, ICA can be used to estimate the mixing matrix *W*. In this work, the Infomax ICA algorithm included in EEGLab [36,37] was employed, which returns the estimate of the mixing matrix *W* that minimizes the mutual information or maximizes the joint entropy between $e_{n}$ and $r_{n}$ in order to make them independent. The implemented algorithm is applied iteratively to overcome the randomness associated with the unsupervised estimation of unmixing matrix with ICA and thus to avoid the arbitrariness in the selection of the recovered sources [38]. The decomposition of instantaneous source interactions achieved by ICA can be applied either separately to each set of residuals *R* obtained solving (Equation 13) for an individual EEG dataset, or collectively to the set of residuals concatenated for each trial and condition. While the first approach would return a different unmixing matrix for each trial and condition, in this work we adopted the second approach which assumes stationarity of the volume conduction effects across trials and conditions. This approach has the advantage of returning a single mixing matrix *W* that is valid for each dataset.

#### 2.2.4. Unmixing

After the decomposition of the residuals of the VAR model of the reduced EEG components by means of ICA, the whole unmixing matrix that relates the source and scalp signals (Equation (Equation 5)) can be obtained combining the CSP filter matrix *C* and the ICA unmixing matrix $W^{-1}$ simply as $U=W^{-1}C$ (Equation (Equation 6)). Then, the (Q×N) data matrix of each set of source time series relevant to the trial *k* of the class $h_{i}$ is reconstructed applying the unmixing matrix to the corresponding set of scalp EEG signals, i.e., $s_{k}^{h_{i}}=Ux_{k}^{h_{i}}$ ($k=1,\ldots ,K_{i};i=1,2$).

#### 2.2.5. VAR Modeling of Source Signals

Once the source signals are estimated, their connectivity structure can be assessed from the VAR representation of Equation (Equation 2). Note that, since such representation is linked to that of Equation (Equation 4), there is no need to identify again a VAR model on the source time series; rather, the VAR parameters $B_{k}$ and $\Sigma_{e}$ can be obtained from the parameters $A_{k}$ and $\Sigma_{r}$ estimated from the reduced EEG components individually for each trial and condition, and from the mixing matrix *W* estimated collectively for all trials and conditions, as $B_{k}=W^{-1}A_{k}W$ and $\Sigma_{e}=W^{-1}\Sigma_{r}W^{-T}$.

#### 2.2.6. Yule-Walker Inverse Solution

The estimated parameters $B_{k}$ and $\Sigma_{e}$ are then exploited to compute the autocorrelation structure of the source signals through the solution of Equation (Equation 7). Specifically, the correlation matrices $\Gamma_{0},\ldots ,\Gamma_{p-1}$ are estimated expressing Equation (Equation 7) in a compact form corresponding to a discrete-time Lyapunov equation (see [30,32] for a detailed treatment). Then, the correlation matrices $\Gamma_{k}$ can be obtained for any arbitrary lag k≥p through iterative application of Equation (Equation 7). In this work, the iteration was repeated up to a lag q=10 to form the covariance and cross-covariance matrices appearing in Equation (Equation 8), which are in turn used for the estimation of information dynamics.

#### 2.2.7. Computation of Information Measures from VAR Sub-Models

The final step of the analysis pipeline is the computation of the measures of information dynamics from the VAR correlation structure of the EEG cortical sources. This step is performed, given a trial *k* of a class $h_{1}$ or $h_{2}$, for each selected pair of driver and target signals $s_{i}$ and $s_{j}$ and grouping the other drivers in the vector signal $s_{k}$, forming the submodels that describe $s_{j,n}$ either from $s_{j,n}^{q}$, or from $s_{j,n}^{q}$ and $s_{k,n}^{q}$, or from $s_{j,n}^{q},s_{i,n}^{q}$ and $s_{k,n}^{q}$. In either case, the prediction error variance (partial variance) relevant to each submodel is identified- without the need to perform the regression- exploiting Equation (Equation 8), where the covariances appearing in the right-hand side of the equation are obtained arranging the correlation matrices found in the previous step. Finally, the partial variances are used to compute the measures of information storage $S_{j}$ (Equation (Equation 9)), information transfer $T_{j}$ (Equation (Equation 9)), and conditional information transfer $T_{i\to j|k}$ (Equation (Equation 9)).

In this work we have also assessed the significant directed links between pairs of nodes in the network of source interactions, which can be defined taking the statistically significant values of the conditional information transfer [14,33]. To do this, we applied the Fisher F-test to compare the two nested regression models used in the computation of $T_{i\to j|k}$, i.e., the full model describing $s_{j,n}$ from $\left[s_{i,n}^{q}s_{j,n}^{q}s_{k,n}^{q}\right]$ and the submodel describing $s_{j,n}$ from $\left[s_{j,n}^{q}s_{k,n}^{q}\right]$. The test statistic is computed from the partial variances $\Sigma_{j|j,k}$ and $\Sigma_{j|i,j,k}$, and is compared with the (1−α)-th percentile of the Fisher distribution with *q* and N−Qq degrees of freedom to reject or confirm the null hypothesis of equal partial variance between the full model and the submodel (here, a significance α=0.05 was assumed).

## 3. Simulation Study

The proposed framework is first tested over simulated EEG signals. In the simulation, two different experimental scenarios (conditions) are designed to reproduce the dynamics of five cortical sources displaying alpha or beta rhythms, which are isolated in the first condition and interact realizing the propagation of alpha waves in the second condition. For each condition we generate, assuming a sampling frequency of 125 Hz, a VAR process of order p=2 composed by Q=5 source processes, in which the diagonal elements of the matrix $B_{k}$ are set to reproduce autonomous oscillations in each scalar source process, and the off-diagonal elements are set to impose specific connectivity patterns (see Ref. [13] for details on the procedure to generate theoretical VAR processes). Specifically, in the first condition autonomous oscillations are set in the alpha band (center frequency f=10 Hz, narrowband) for $s_{1}$, $s_{4}$, and $s_{5}$, and in the beta band (center frequency f=23 Hz, broadband) for $s_{2}$ and $s_{3}$; all off-diagonal elements of $B_{k}$ are set to 0 in order to reproduce absence of connectivity. In the second condition, the autonomous oscillations are set in the alpha band for $s_{1}$ and in the beta band for $s_{2}$, $s_{3}$, $s_{4}$, and $s_{5}$; in this case causal effects are imposed along the chain $s_{1}\to s_{2}\to s_{3}\to s_{4}\to s_{5}$, by setting $B_{1}\left(2,1\right)=B_{1}\left(3,2\right)=B_{1}\left(4,3\right)=B_{1}\left(5,4\right)=0.5$ in order to reproduce the propagation of the alpha rhythm from $s_{1}$ to $s_{5}$. Note that, given the theoretical values imposed for the VAR parameters, the exact values of the measures of information dynamics can be computed, and are used here as the ground truth for evaluating the performance of the algorithms. Then, 10 realizations (trials) of the source time series of length N=1000 samples are obtained for each condition feeding the simulated VAR model with non-gaussian innovations; the non-gaussian noise is generated applying the nonlinear transformation $e_{n}=sign\left(w_{n}\right){\left|w_{n}\right|}^{q}$ to a gaussian white noise $w_{n}$ (q∈[1.2,2] to yield super-Gaussian distributions [39]). The source time series are then multiplied with a mixing matrix *M* to obtain the simulated scalp EEG signals (Equation (Equation 1)). We choose a square mixing matrix (D=Q=5) to allow the unmixing matrix to be full rank and thus invertible; the entries of *M* are chosen to simulate the volume conduction effect of a source over two or three electrodes:(14)$$M=\left[\begin{array}{ccccc} 1 & 0.5 & 0 & 0 & 0\\ 0.5 & 1 & 0.5 & 0 & 0\\ 0 & 0.5 & 1 & 0.5 & 0\\ 0 & 0 & 0.5 & 1 & 0.5\\ 0 & 0 & 0 & 0.5 & 1 \end{array}\right].$$

The analysis of information dynamics is first computed at the scalp level, i.e., identifying the VAR model directly on the D=5 simulated EEG signals and computing the information measures from the estimated VAR parameters (steps (v, vi, vii) of the analysis, see Figure 1). Then, the analysis is repeated applying the whole framework, i.e., applying in sequence CSP, VAR identification, ICA, reconstruction of the source model and computation of the information measures according to the whole pipeline depicted in Figure 1.

Results are shown in Figure 2, which depicts the values of the information measures computed from the simulated scalp signals (a,b) and from the reconstructed source signals (c,d) in the first condition of non-interacting sources (a,c) and in the second condition with directed connectivity among sources (b,d). The analysis on the scalp signals shows that, even in the first condition with absence of causal interactions between the sources, the total and conditional information transfer assessed for the simulated EEG signals is substantial (high values of $T_{j}$ and $T_{i\to j|k}$ in Figure 2a), providing a false indication of directed connectivity. On the contrary, when the proposed CSPVARICA approach is applied, the lack of connectivity is clearly documented by the null values of the information transfer to each source and of the directed transfer between pairs of sources (null values of $T_{j}$ and $T_{i\to j|k}$ in Figure 2c). In the second condition, the scalp analysis remains confounding as the values of information transfer are scattered across the connectivity matrix (Figure 2b), while the source analysis reveals the ability of the proposed framework to elicit the directional interactions that are imposed between the sources (i.e., significant directed connectivity from $s_{1}$ to $s_{2}$, from $s_{2}$ to $s_{3}$, from $s_{3}$ to $s_{4}$, and from $s_{4}$ to $s_{5}$, highlighted by the high values of $T_{i\to j|k}$, Figure 2d). Remarkably, the distributions across trials of information storage and information transfer are well aligned with the theoretical values resulting from the true VAR source parameters (filled symbols in Figure 2c,d), thus documenting the accuracy of the reconstruction.

## 4. Application to EEG Signals in Epileptic Children

### 4.1. Experimental Protocol, Data Preprocessing and Statistical Analysis

EEG signals were acquired during night monitoring from patients suffering from focal or generalized epileptic seizures. The study group consisted of 15 patients (3–25 years, 11 males and 4 females) in whom episodes of central or temporal focal seizures were observed, and 5 patients (2–13 years, 4 females and 1 male) with episodes of generalized epilepsy. Further information about the seizure type and onset are available in Table 1.

Antiepileptic medications were administered in therapeutic dosages to the patients according to their diagnosis, i.e., (i) children with focal seizures: carbamazepine, lamotrigine, lacosamide, topiramate, valproic acid; (ii) children with generalized seizures: valproic acid, lamotrigine, topiramate, and vigabatrin [40].

In total, 45 episodes of focal seizures and 22 episodes of generalized seizures were monitored. In patients with focal seizures, an average of 2.5 seizures per patient was analyzed (range: 1–8). In patients with generalized seizures, the average number of seizures analyzed per patient was 3.8 (the range was 1–3 in four patients, while the fifth exhibited 13 seizures). Signals were recorded at a sampling rate of 250 Hz by scalp electrodes placed according to the International 10/20 monopolar scheme, with the reference electrode placed on the ipsilateral ear. During the study, the acquisition of EEG signals was accompanied by continuous video monitoring to identify clinical events. In order to define the start and stop time of each seizure, the episode onsets were labeled in the EEG by qualified neurologists based on visual analysis of EEG data and video recordings as the events of the start of clinical appearance of an actual seizure. Figure 3 shows an example of EEG signals containing the start of focal (Figure 3a) and generalized (Figure 3b) seizures marked with dashed lines.

Preprocessing consisted in downsampling each signal to 125 Hz, followed by application of a bandpass filter (zero-phase IIR Butterworth filter) with cutoff frequencies at 0.5 Hz and 42 Hz. Only stationary EEG trials without noticeable artifacts were manually selected for the analysis. The EEG time series were then normalized to zero mean and unit variance before calculating information theoretic measures. More details on data acquisition and pre-processing are given in previous studies [40,41].

In order to investigate the reorganization of the networks of EEG connectivity possibly occurring before and after epileptic seizures, three time windows were considered for the analysis: base, starting 10 s before and ending 5 s before a se izure; pre, starting 5 s before a seizure and ending at the moment of sei zure onset; post, starting immediately after seizure termination and ending after 5 s. Periods during seizure were excluded from the analysis, due to the frequent observation of highly nonstationary and noisy EEG signals that could not meet the requirements for VAR analysis. After preprocessing and segmentation, the measures of information dynamics were computed on the cortical source signals reconstructed from the EEG according to the whole analysis procedure described in Section 2.2. The analysis was performed, separately for the trials of focal and generalized seizures, for each pair of classes to which CSP can be applied (i.e., base vs. pre, pre vs. post, and base vs. post). Moreover, to allow a comparison between EEG networks assessed at the level of scalp sensors and cortical sources, computation of the information measures was performed also directly on the scalp signals, i.e., executing the steps (v, vi, vii) of the analysis (see Section 2.2 and Figure 1) on the pre-processed EEG signals.

Since each subject had different number of seizures and given that only stationary and artifact-free recordings were used for the analysis, every subject contributed to the experimental data set with a different number of trials. To address this issue as regards the statistical analysis, the obtained values of information storage, total information transfer and conditional information transfer were averaged across all trials from each subject, thus obtaining only one value per subject. Moreover, to assess the variability across trials of each information measure, the percentage mean absolute deviation from the mean (MAD) was calculated as follows [42]:(15)$$MAD=\frac{100}{N_{subj}}\sum_{subj=1}^{N_{subj}}\left[\frac{1}{K_{h,subj}}\sum_{k=1}^{K_{h,subj}}\frac{|M_{k,subj}-\mu_{M,subj}|}{\mu_{M,subj}}\right]$$
where $M_{k,subj}$ is the measure computed for the trial *k* of the patient subj during the condition *h*, $\mu_{K,subj}$ is the mean of the considered measure across trials of the patient subj, $K_{h,subj}$ is the number of trials of the condition for that patient, and $N_{subj}$ the number of patients.

Afterwards, the distributions across subjects obtained for each information measure were analyzed to infer statistical differences between pairs of classes (i.e., base/pre, base/post, pre/post). Statistical analysis was carried out using the nonparametric two-sided Wilcoxon rank-sum test. The obtained *p*-values were not corrected for multiple comparisons, but this does not affect the analysis given the small number of comparisons and, with regard to source signals, since the unmixing is specific for each pair of classes compared and thus results in different distributions.

Moreover, the effect size was evaluated according to Cohen’s *d* formula [43]:(16)$$d=\frac{\mu_{1}-\mu_{2}}{\sigma }$$
being $\mu_{1}$ and $\mu_{2}$ the means of the two distributions taken into account and σ the pooled standard deviation calculated as [43]:(17)$$\sigma =\sqrt{\frac{\left(n_{1}-1\right)\sigma_{1}^{2}+\left(n_{2}-1\right)\sigma_{2}^{2}}{n_{1}+n_{2}-2}}$$
with $n_{1}$ and $n_{2}$ the sample size and $\sigma_{1}^{2}$ and $\sigma_{2}^{2}$ the variance of each distribution. The effect size values obtained as above described were then interpreted according to the rules of thumb defined by Cohen [43] expanded through the descriptions given by Sawilowsky [44], to get a classification of the effect as “very small”, “small”, “medium”, “large”, “very large” or “huge”.

### 4.2. Results

#### 4.2.1. EEG Model Identification and Validation

The first step of the approach proposed to model EEG source connectivity is the application of CSP on the pre-processed EEG signals measured in two classes (here, base-pre, pre-post, or base-post). As reported in Section 2.2.1, CSP finds the *Q* EEG components that maximize the difference in variance between the two analyzed classes on the basis of the Riemannian distance between the two class-related covariance matrices (Equations (Equation 11) and (Equation 12)). With this approach, we found that Q=10 CSP spatial filters (corresponding to 10 eigenvalues) were sufficient for describing 93% of the total distance between the covariance matrices in all the three paired comparisons performed both for focal and generalized seizures. The results of this computation for the comparison between the classes pre and post of the EEG signals relevant to focal seizures are reported in Figure 4, showing that Q=10 CSP spatial filters cover 97% of the total distance. Repeating the same analysis for different pairs of classes, we found that the use of 10 filters resulted in a contribution to the total squared Riemaniann distance equal to 93% and 95% when comparing the base-pre and base-post trials of the focal seizures, and equal to 94%, 94% and 97% when comparing the base-pre, pre-post and base-post trials of the generalized seizures.

After CSP, the time-lagged connectivity structure of the reduced EEG components was estimated identifying VAR models for each individual trial and condition. Estimation of the optimal model order, performed through the SBC criterion, resulted in orders ranging from 25 to 30 for both source and scalp EEG signals with focal and generalized seizures. The subsequent test for whiteness of the model residuals, performed using the Li-McLeod test as Portmanteau statistics, indicated uncorrelation of the residuals as the hypothesis of white residuals was not rejected in any of the analyzed trials.

#### 4.2.2. Information Dynamics from EEG Scalp Signals

In our first analysis we computed the measures of information dynamics on the 19 preprocessed scalp EEG signals, through VAR modeling performed without application of CSP and ICA. The measures of information storage $S_{j}$ and total information transfer $T_{j}$ computed for each target EEG signal (j=1,…,19), as well as the measure of conditional information transfer $T_{i\to j|k}$ computed for each possible pair of source and target EEG signal (i,j=1,…19,i≠j) were averaged to obtain a single value for each trial. Moreover, the number of directed links (pairs of channels i→j) resulting as statistically significant according to the F-test, denoted as $N_{i\to j|k}$, was also computed. The values obtained for each of these measures were further averaged across all trials for the same subject in any given condition.

The resulting distributions are reported in Figure 5 for generalized seizures and in Figure 6 for focal seizures. Comparing each measure between pairs of conditions in the generalized seizure group (Figure 5), we found a statistically significant decrease of the average amounts of information stored in the network nodes (p< 0.05), total information transferred to the nodes (p< 0.05), conditional information transfer between node pairs (p< 0.01), and of the number of significant links, moving from base to post and from pre to post, but not moving from base to pre (*p*-values not corrected for multiple comparisons are reported). The analyses carried out studying the average percentage Mean Absolute Deviation calculated on all the information measures reported higher MAD values for the post condition. In detail, for scalp EEG signals, MAD was around 9.7% in base, 13.5% in pre, 38.1% in post for focal seizures, 7.9% in both base and pre and 90.0% in post for generalized seizures. For source EEG signals, MAD was around 14.5% in base, 17.6% in pre, 41.3% in post for focal seizures, 9.5% in base, 11.8% in pre and 92.3% in post for generalized seizures. Table 2 and Table 3 report the values of Cohen’s *d* effect size calculated for the measures of information storage $S_{j}$, information transfer $T_{j}$, conditional information transfer $T_{i,j\to k}$ and number of statistically significant network links $N_{i,j\to k}$ computed from the scalp signals in patients with generalized and focal seizures. The results generally reflected those relevant to the statistical analysis. In detail, for generalized seizures a small effect size was reported for $S_{j}$, $T_{j}$, $T_{i,j\to k}$ in the comparison base-pre, while a huge effect size was found comparing base-post and especially pre-post. In the case of focal seizures, the effect size was small-to-medium for base-pre, medium for pre-post and large for base-post (apart from $N_{i,j\to k}$).

Overall, these results suggest a remodulation of the information dynamics of the scalp EEG signals occurring after generalized seizures. This effect seems peculiar of generalized seizures, since in the analysis of the scalp EEG signals related to focal seizures, no significant differences were found for any of the information measures and for each comparison between pairs of conditions (Figure 6).

#### 4.2.3. Information Dynamics from EEG Sources

Then, the analysis of information dynamics was carried out at the level of the EEG sources, exploiting the full framework proposed in this study whereby CSP, VAR modeling and ICA are applied to assess reduced EEG components and model simultaneously instantaneous volume conduction and time-lagged source interaction effects. The analysis was performed again computing the information storage $S_{j}$, the total and conditional information transfer $T_{j}$ and $T_{i\to j|k}$, as well as the number of statistically significant directed links $N_{i\to j|k}$, averaged first over all Q=10 nodes or Q(Q−1)=90 connection pairs in the source network to get a single measure per trial, and then over all trials to get a single measure per subject. Moreover, since in this case the unmixing is specific for each pair of classes compared (base-pre, pre-post and base-post), the estimated source time series and therefore the estimated values of information dynamics are generally different in the same class when CSP is applied to a different pair of classes. For this reason, the distribution of values is shown two times per condition in Figure 7 and Figure 8, grouping pairs of distributions on the basis of CSP analysis. The results of the analysis of information dynamics for the EEG sources are presented in Figure 7 and Figure 8 for generalized and focal seizures, respectively. In general, the statistical tests on the measures computed at the source level indicated overall lower *p*-values compared to the analysis performed at the scalp level.

In the case of generalized seizures, the results were comparable to those obtained for the scalp signals. Indeed, as seen in Figure 7, the information storage, the total and conditional information transfer, and the number of network links with significant directed interaction exhibited a tendency to decrease when comparing the conditions pre-post and base-post. Again, this suggests a general tendency of the information dynamics to be blunted after generalized epileptic seizures compared to the periods preceding the ictal events. This tendency is clearly confirmed by the very large or huge effect size of the changes of $S_{j}$, $T_{j}$ and $T_{i\to j|k}$ reported in Table 4 for the comparisons pre-post and base-post.

On the other hand, the analysis carried out on EEG source dynamics in the group of patients with focal seizures (Figure 8) suggested a behavior not revealed at the scalp level, i.e., the tendency of the information transfer between pairs of nodes of the source network to increase before seizure and return to interictal values after the ictal events. This tendency was documented by the *p*-values close to statistical significance (p∼ 0.06–0.1) of the differences both in the total and in the conditional information transfer observed moving from base to pre (increase of $T_{j}$ and $T_{i\to j|k}$), and moving from pre to post (decrease of $T_{j}$ and $T_{i\to j|k}$). The tendency was confirmed by the Cohen’s *d* analysis, as reported in Table 5 showing d∼−0.5 (medium-to-large effect size according to [43,44]) in the comparison base-pre and d∼0.8 (large effect size) in the comparison pre-post, and these trends were not observed in the analysis of scalp signals. The changes in the information transfer were observed in the absence of significant changes of the number of nonzero directed links $N_{i\to j|k}$, suggesting that the variations of the information flow occur over a similar topological structure of the EEG source networks. Moreover, the lack of statistical significance in the comparison base-post suggests that, after the increase in the preictal period, the three indexes return to baseline values after the seizure. As regards the measure $S_{j}$, the absence of significant changes across conditions documents a behavior for the information storage in the EEG sources similar to that observed for the scalp signals.

## 5. Discussion

The analysis of information dynamics, and in particular of the measures of information storage and information transfer, plays a fundamental role in neural computation, especially with regard to the network representation of the electrical activity of the brain [45,46]. The estimation of information dynamics, in particular information transfer, from scalp EEG signals is severely complicated by volume conduction effects which may dominate the transfer of information across EEG sensors and therefore blur the detection of meaningful patterns of information flow [21,22,47]. While solutions to this problem have been proposed which keep working with scalp signals and try compensating volume conduction [45,47,48], a more intuitive approach is to work on the source signals reconstructed from the sensor activity since it has been demonstrated that the source-based network representation constitutes a better approximation of the unknown true network structure [17]. In this work we pursue this approach, integrating it into existing approaches for the linear parametric representation of the dynamics of cortical sources and of their mixing to produce scalp EEG signals [27,28]. The resulting framework constitutes, to the best of our knowledge, the first attempt to access the information dynamics of EEG sources through simultaneous modeling of time-lagged source interactions and instantaneous mixing effects. This is accomplished combining CSP which allows reducing the dimensionality of EEG signals while taking into account nearly the same amount of information in terms of variability, VAR for combined source and scalp EEG modeling, and ICA for obtaining instantaneously independent cortical sources.

The application of the proposed framework to a neural disorder characterized by abrupt changes in the dynamical state of the brain, i.e., epileptic seizures, led us to highlight the usefulness of source analysis of information dynamics in comparison to scalp analysis. In particular, we found similar information-theoretic measures computed from scalp and source signals in the case of generalized seizures, while differences between the two approaches were revealed for focal seizures. The different results are likely due to the characteristics of brain activity for different seizure types [49]. In the case of generalized seizures, the entire brain volume is entangled in the seizure activity, which is thus observed everywhere in the brain; as a consequence, one may expect that volume conduction does not alter substantially the patterns of interaction among sources, which should be observed similarly at the level of the scalp. On the contrary, in focal epileptic seizures, the activity is more localized in a brain subvolume nearby the seizure origin, and volume conduction causes its spreading to other locations and appearance at scalp sensors far away from the actual seizure focus location; this is expected to have an effect on how connectivity patterns are detected at the scalp compared with the source level. In fact, analyzing focal seizure episodes we did not detect any remarkable variation across conditions in the measures of information dynamics computed at the scalp, while the changes in the information transfer were noticed when considering the EEG sources. Remarkably, the profiles across conditions of the measure of information storage were similar when analyzing scalp and sensor signals, also for the focal seizures. These results are in line with previous findings [47] evidencing that, in response to an action which evokes localized occipital brain reactions like the eyes closure, information storage is informative also when computed at the scalp level, likely because of its univariate nature, while information transfer is blurred to uniform values due to the adverse effects of volume conduction on connectivity measures. To highlight the importance of a distinction between different type of seizures, we remark that differences have been evidenced also in relation to autonomic nervous regulation, e.g., probed by measures (including information-theoretic ones) of heart rate variability computed before and after focal and generalized seizures [40].

Although obtained from a small number of patients, the analysis of generalized seizures evidenced clear patterns of alteration of the measures of information dynamics when comparing the epochs preceding and following the ictal episodes. In particular, both the amounts of information stored in the EEG signals and in the reconstructed sources, and the amounts of information transferred within the scalp network and within the source network, decreased significantly moving from the pre to the post window. This finding suggests that a reorganization of the brain network underlying spontaneous brain activity in non-ictal conditions takes place after the ictal episodes, with decreased ability of the network nodes to actively store information and to send information between each other. On the other hand, the absence of significant differences in the comparison between the base and pre conditions suggests the lack of predictive value for the measures of information dynamics, at least regarding the onset of generalized seizures.

As regards focal seizures, the analysis performed through our approach at the level of the cortical sources evidenced changes of directed connectivity within the source signals in both the comparisons base/pre and pre/post, with the conditional and particularly the total information transferred to the network nodes increasing just before the seizure and returning to basal values just after the seizure. Such trends suggest alterations in the connectivity of the source EEG network occurring before the onset of focal epileptic seizures. The use of information-theoretic measures as an indicator of phase transitions has been demonstrated using classic spin model in physics [50], as well as in financial systems [51] and in the brain: recent studies have shown that the maximal amount of information transfer among units is representative of the critical state on a brain network [52], and that synergy measures derived from information transfer in a multivariate fashion peak before the transition from disordered to ordered phases [53]. In this context, the use of measures of information transfer like those presented in this study, possibly integrated with multivariate measures obtained in the context of information dynamics [54], may have important implications in seizure prediction. If computed within the EEG source network, such measures are indeed good candidate tools to identify peaks of information flow located away from the time of transition between non-ictal and ictal brain activity, being thus predictive of the transition itself.

A limitation of our analysis consists in the fact that the time windows chosen as baseline might not necessarily be free of epileptic activity and already reflect an altered state of the brain, since often a seizure is preceded by long periods of seizure-readiness of the brain. This may explain the little difference found between base and pre conditions. Future studies on more complete databases will analyze interictal periods where the difference with the seconds preceding the seizure onset is expected to be larger. Another potential limitation of the present study is that the proposed methodology is tested on a quite restricted database containing EEG acquired on children of different genders and type of seizure, and substantial variability in age. The reported variability across episodes of the information measures, especially in the post-ictal phase of generalized seizures, also suggests that larger and more homogeneous databases should be employed in future works. The high variability after the seizure is likely related to the difficulty in establishing clearly and objectively when the seizure is over, as seizures can terminate across brain areas via a variety of spatiotemporal patterns [55]. These aspects might jeopardize the acceptance of the methodology and question its generalization ability in a clinical setting. In addition, using the EEG signals from subjects with wide age range one might expect developmental changes in characteristics (e.g., seizure semiology [56] or interictal discharges [57]) that should be evaluated separately in narrower age groups during thorough studies, including the developmental characteristics of the information dynamics measures. On the other hand, comparable group sizes are used in other works on the dynamics of epilepsy, such as in [58] for recurrence quantification analysis, or in [59] for studying heart rate variability as an indicator of autonomic dysregulation. While application on larger benchmarking datasets is a primary focus of future work, the aim of this paper was to present the proposed framework addressing all technical aspects and its evaluation in a simulation setting, while also showing its potential in a clinical application. Besides, the proposed framework might be further developed and improved, as some open questions emerged during the research. First of all, since the present study demonstrated promising results of the application of selected information-theoretic measures to the EEG sources, other measures should be added to the framework, to enable studying of brain activity characteristics in two conditions. Exploiting both available and new measures requires the investigation of their robustness with respect of the initial data for computation of EEG sources, i.e., duration of the EEG recordings, number of seizure instances per class and per subject, stationarity of the signal, etc. This should unveil any existing limitations in the applicability of the proposed approach to study rare EEG phenomena, or dealing with small number of subjects, which is often the case in EEG research, as well as evaluation of the results of the analysis in subject-specific vs. non-subject specific settings. A further important extension of the proposed framework would be adding the time and spatial dimensions to the description of the information-theoretic characteristics of brain activity. Proper assessment of time-varying measures of EEG sources requires the study of the VAR model fitting and finding the unmixing matrices for different durations and time positions of trials in classes under comparison to find optimal combination of these settings to compute the informative EEG sources. Also, since CSP provides the spatial distribution of activity patterns, the proposed framework could be extended to account the localization of sources in combination with assessing their entropy and connectivity characteristics. Interpretation of the CSP spatial characteristics and relation of those with the actual localization over the scalp and in the brain tissue volume might be an important advancement of the framework, as it would open the way to the use of directed information measures for the localization of the drivers of epileptiform activity [60].

## 6. Conclusions

Characterization of brain connectivity and assessment of information dynamics is feasible in the network of EEG source activations rather than in scalp EEG signals using the proposed framework. The benefits of a combined scalp-source modeling approach performing dimensionality reduction, description of volume conduction effects and parametric analysis of information measures are evidenced in this study first showing the ability to reconstruct imposed patterns of information dynamics in a simulation setting, and then reporting the identification of distinct behaviors associated to generalized and focal epileptic seizures. Analysis frameworks like the one presented in this work may help the unification and standardization of views concerning the epileptic seizures occurrence and spreading, and in perspective might be used to provide new features to seizure detection or prediction algorithms.

## Figures and Tables

**Figure 1 brainsci-10-00657-f001:**
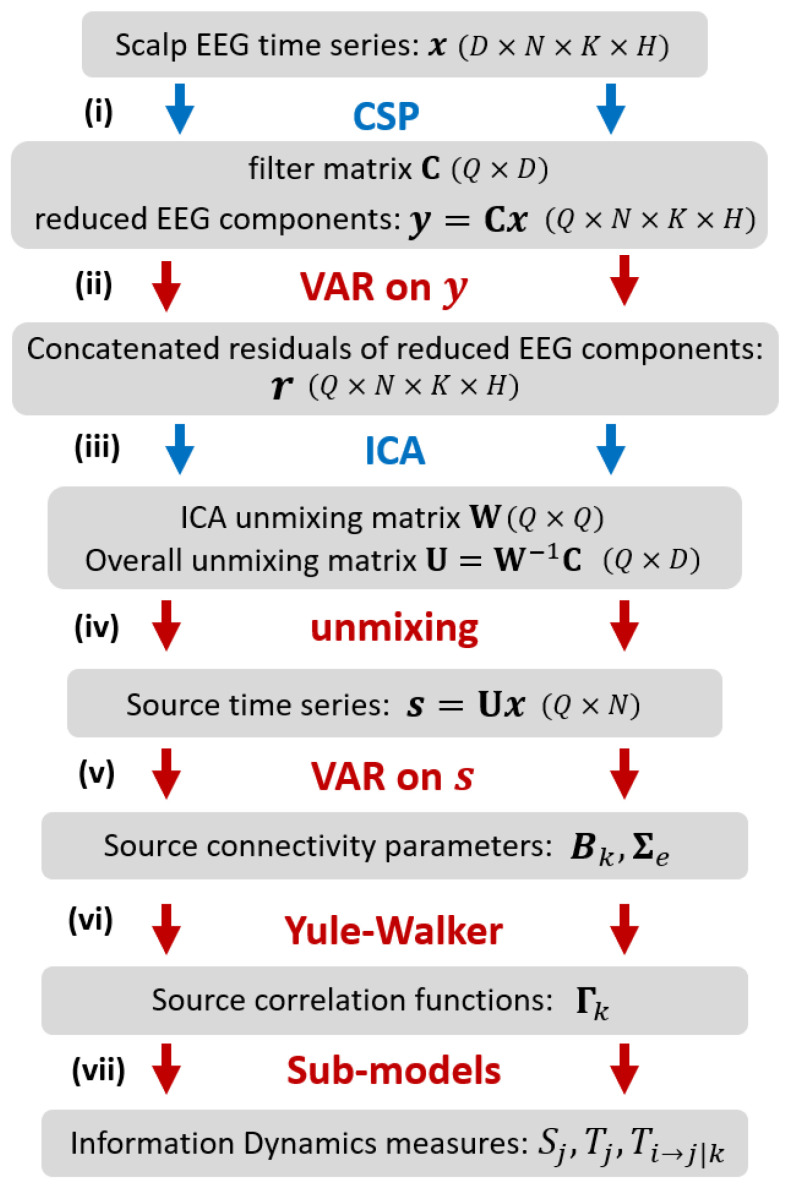
Description of the methodology adopted in this work to assess information dynamics of EEG cortical sources. The EEG signals (in the number of *D* signals, each with length of *N* samples, collected over *K* trials during *H* conditions) are reduced in dimension by CSP (from *D* signals to *Q* components), and their time-lagged and instantaneous interactions are described respectively by a VAR model and by ICA; then, source dynamics are reconstructed via unmixing of the scalp signals, their interactions are retrieved through VAR modeling and solution of Yule-Walker equations, and information measures are computed from sub-models relevant to specific sources. In the figure, blue and red arrows refer to analysis steps performed on the whole dataset and on single trials, respectively. More details on the analysis are provided in the text.

**Figure 2 brainsci-10-00657-f002:**
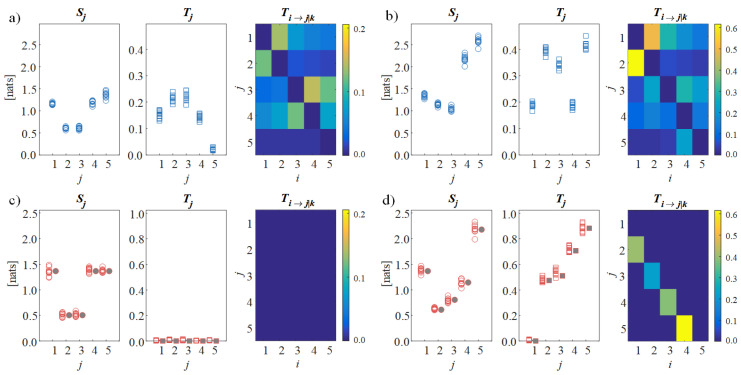
Estimation of information dynamics for the simulated scenarios. Plots depict the distributions across 10 trials of the information storage ($S_{j}$, circles) and the total information transfer ($T_{j}$, squares), as well as the color-coded median values of the conditional information transfer ($T_{i,j\to k}$, connectivity matrices) computed for the simulated scalp signals (**a**,**b**) and for the cortical sources reconstructed using the proposed approach (**c**,**d**) in the first condition of absence of connectivity (**a**,**c**) and in the second condition with unidirectional source propagation (**b**,**d**). The filled symbols in (**c**,**d**) indicate the exact values of the information stored in each cortical source and the total information transferred to it.

**Figure 3 brainsci-10-00657-f003:**
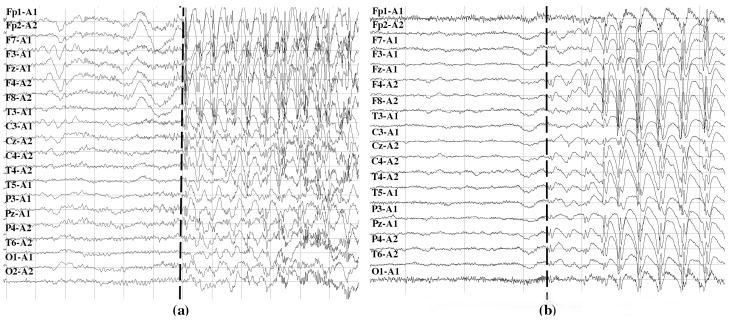
Examples of EEG signals recorded with monopolar montage with the reference electrode placed on the ipsilateral ear at the onset of focal (**a**) and generalized (**b**) seizures. In the two cases, seizure onsets are marked with dashed lines. It can be noticed that the epileptiform discharges begin in the left frontotemporal region with later bilateral synchronization in the right frontal region in (**a**), and in all brain areas at the same time in (**b**).

**Figure 4 brainsci-10-00657-f004:**
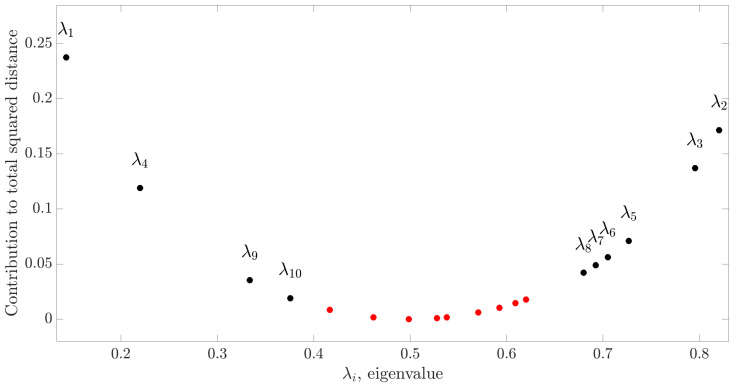
Contribution to the total squared Riemannian distance provided by the eigenvalues $\lambda_{j}$ corresponding to each spatial filter $c_{j}$ (j=1,…,19) of the CSP matrix obtained for the generalized seizures trials classified as base and pre. Black and red dots identify the selected spatial filters and the discarded filters, respectively.

**Figure 5 brainsci-10-00657-f005:**
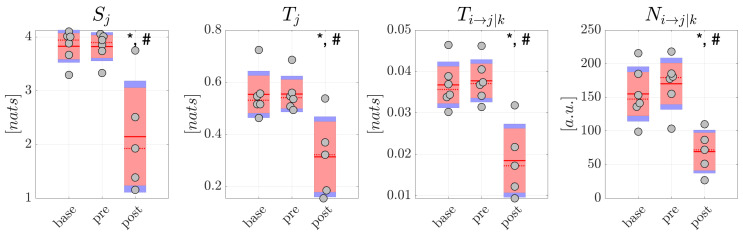
Information dynamics of scalp EEG signals measured during generalized seizures. Panels report the boxplot distribution and individual subject values (average over network nodes and trials) of the information storage $S_{j}$, total information transfer $T_{j}$, conditional information transfer $T_{i\to j|k}$, and number of significant links $N_{i\to j|k}$ in the three analyzed conditions base, pre, post. Statistically significant differences between pairs of distributions: *, *p* < 0.05 base vs. post; #, *p* < 0.05 pre vs. post.

**Figure 6 brainsci-10-00657-f006:**
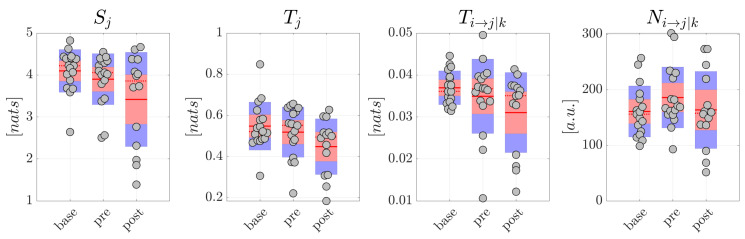
Information dynamics of scalp EEG signals measured during focal seizures. Plots and symbols are as in Figure 5.

**Figure 7 brainsci-10-00657-f007:**
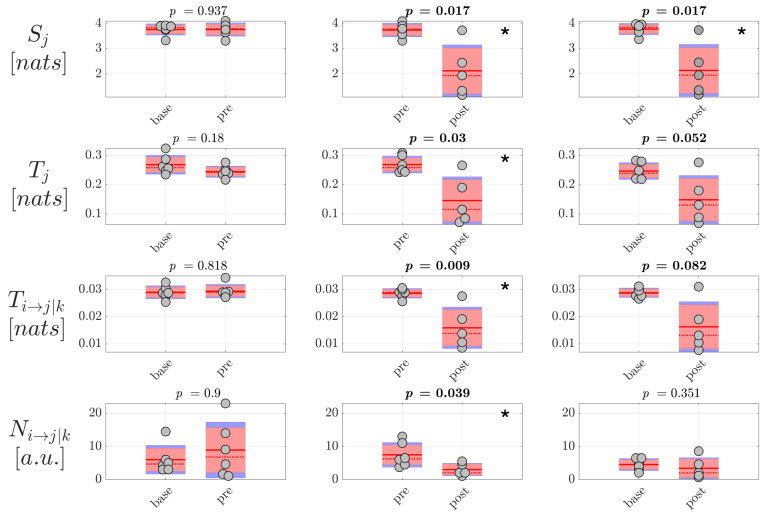
Information dynamics of source EEG signals measured during generalized seizures. Panels report the boxplot distribution and individual subject values (average over network nodes and trials) of the information storage $S_{j}$, total information transfer $T_{j}$, conditional information transfer $T_{i\to j|k}$, and number of significant links $N_{i\to j|k}$ in the three analyzed conditions base, pre, post. Statistically significant difference between pairs of distributions (*p* < 0.05) are marked with *; *p*-values < 0.1 are written in bold.

**Figure 8 brainsci-10-00657-f008:**
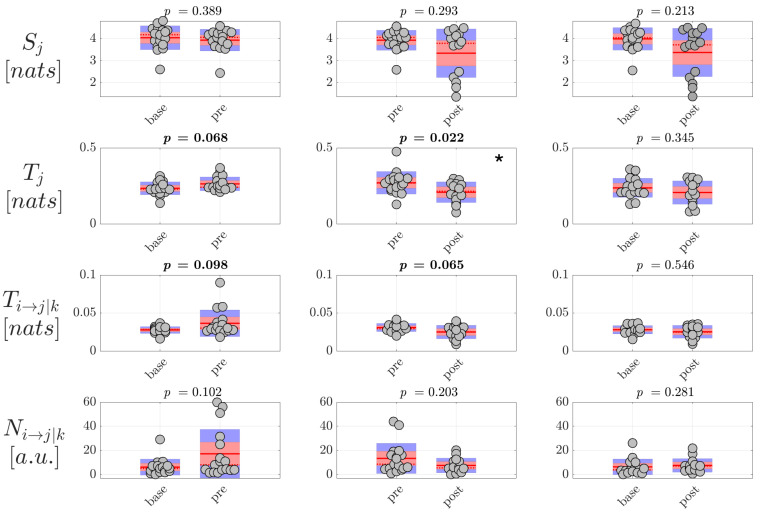
Information dynamics of source EEG signals measured during focal seizures. Plots and symbols are as in Figure 7.

**Table 1 brainsci-10-00657-t001:** Information about seizures type and onset.

N. Subjects	Group	Seizure Type
1	Focal	Left temporal and frontal-temporal
1	Focal	Right frontal-temporal
1	Focal	Central
1	Focal	Right frontal-temporal and frontal-central
1	Focal	Right central
1	Focal	Left frontal-temporal with secondary bilateral synchronization
2	Focal	Right frontal
1	Focal	Bifocal asynchronous epileptiform activity in the right frontal-temporal, left central-parietal and parietal
1	Focal	Central-parietal
1	Focal	Left central-temporal
4	Focal	Unknown
1	Generalized	Idiopathic generalized seizures with typical absences
1	Generalized	Vesta Lennox–Gastaut syndrome, asymmetric infantile spasms
1	Generalized	Absence seizures
1	Generalized	Generalized tonic, myoclonic seizures
1	Generalized	Jeavons syndrome, myoclonic seizures

**Table 2 brainsci-10-00657-t002:** Cohen’s *d* effect size-Generalized seizures-Scalp signals.

	Base−Pre	Pre−Post	Base−Post
**$S_{j}$**	0.022	2.330	2.314
**$T_{j}$**	−0.016	2.097	1.954
**$T_{i\to j|k}$**	−0.183	2.739	2.538
**$N_{i\to j|k}$**	−0.383	2.827	2.314

**Table 3 brainsci-10-00657-t003:** Cohen’s *d* effect size-Focal seizures-Scalp signals.

	Base−Pre	Pre−Post	Base−Post
**$S_{j}$**	0.354	0.550	0.809
**$T_{j}$**	0.246	0.551	0.797
**$T_{i\to j|k}$**	0.293	0.426	0.838
**$N_{i\to j|k}$**	−0.497	0.361	−0.049

**Table 4 brainsci-10-00657-t004:** Cohen’s *d* effect size—Generalized seizures—Source signals.

	Base−Pre	Pre−Post	Base−Post
**$S_{j}$**	0.155	2.233	2.315
**$T_{j}$**	−0.763	2.023	1.652
**$T_{i\to j|k}$**	0.106	1.960	1.990
**$N_{i\to j|k}$**	−1.002	1.155	0.438

**Table 5 brainsci-10-00657-t005:** Cohen’s *d* effect size—Focal seizures—Source signals.

	Base−Pre	Pre−Post	Base−Post
**$S_{j}$**	0.123	0.684	0.741
**$T_{j}$**	−0.487	0.853	0.440
**$T_{i\to j|k}$**	−0.548	0.817	0.389
**$N_{i\to j|k}$**	−0.709	0.588	−0.203

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
