# Peer review of "A Framework to Assess the Information Dynamics of Source EEG Activity and Its Application to Epileptic Brain Networks"

_brainsci, 2020, doi:10.3390/brainsci10090657_

Round 1
Reviewer 1 Report
I appreciate the detailed answers of the authors.
Finally, like 'Equation', can you revise all 'Fig.' to 'Figure.'?
Reviewer 2 Report
Thank you for asking me to re-review this manuscript. The authors have been very responsive to my previous comments. I have no further concerns.
Reviewer 3 Report
This is an interesting study about the differential results obtained by employing connectivity analysis with and without transformation to source level.
Abstract:
line 12: do you mean "a significant decrease from pre-ictal to post-ictal periods of the information stored..."
line 15: during focal epilepsy does not make sense - do you mean during focal seizures?
line 17: I assume this should also be focal seizures, and not focal epilepsy.
line 21: "and their potential relevance ..."
Introduction:
Line 28: "Characterization of the brain function is usually conducted through the analysis of functional and effective connectivity," I would not say this is the usual way, there are numerous other ways! you can write "is often conducted..."
Line 29: I would be cautious to say they reflect causal relationships - they are interpreted as causal relationships!
Concept:
This sounds to me like a methodological paper, not an original paper. The results presented are not quite surprising, but the method may be new. So the abstract should be aligned to that, emphasizing that the scope of the paper is to introduce a new method.
Figure 1: This should be provided as an equation, not a figure, and the variables should be defined. Like this it is impossible to assess what the authors mean and the figure is not of much use.
Section 3.1:
Comparing central+temporal with generalized epilepsy is maybe a bit
I do not fully agree that the time during a seizure cannot be analyzed. This has been done by several research groups with interesting findings, e.g.
Epileptic seizures as condensed sleep: an analysis of network dynamics from electroencephalogram signals
HEIDEMARIE GAST1*, MARKUS MÜLLER2,3*, CHRISTIAN RUMMEL4, CORINNE ROTH1, JOHANNES MATHIS1, KASPAR SCHINDLER1
and CLAUDIO L. BASSETTI1
What has been selected as "base" might not necessarily be free of epileptic activity, as a seizure is often preceded by minutes of seizure-readiness of the brain. Moreover, the seizure onset can be arbitrary, such that little difference can be expected between "base" and "pre" conditions. This is also reflected by the results, where base- vs. pre- look very similar.
page 9 line 278 typo: averaged
It would be good to know how many "trials" i.e. seizures there were per patient. The variance across seizures can be considerable, so a characterization of that variance would be needed.
Section 3.2 Results:
It is not clear whether the authors corrected for multiple comparisons, as they compared different conditions by various biomarkers.
line 337 typo distributions
line 386 typo observed
Discussion: The conclusions on the lack of of pre- to base differences are not valid, given the choice of the time-windows. Base should be much more distant in time to the beginning of the seizure.
line 456 typo (toi)
More details on the patients are needed:
Age, gender, onset of epilepsy, medication...
The introduction and discussion should include more references on the two perspectives on source-level analysis: One view is that this is the only way this can be done, because of volume conduction. The other view is however considering all the disadvantages, i.e. the pre-defined number of sources depending on the number of sensors, which is not plausible, nor realistic, and the mathematical constraints and assumptions woven into that approach, which do not necessarily reflect reality.
I recommend reading and citing some critical authors, e.g.
Evolving networks in the human epileptic brain
Klaus Lehnertza,b,c,∗, Gerrit Ansmanna,b,c, Stephan Bialonskia,b,c, Henning Dicktena,b,c,
Christian Geiera,b, Stephan Porza,
The elusive concept of brain connectivity Barry Horwitz*
And I do also have the feeling that prior work that compared source-level with scalp-level analysis was not cited conclusively:
https://www.nature.com/articles/s41598-018-30869-w
Round 2
Reviewer 3 Report
I highly appreciate the thorough revision of the authors and the detailed response, very well done!
The issue about focal and generalized was not addressed sufficiently:
You can have focal or generalized epilepsy.
But if you have focal epilepsy, you can still have generalized seizures.
Therefore I suggest not talking about epilepsy in the abstract, but about seizures (line 16) unless you really mean patients with focal epilepsy and ANY type of seizures. But then, the question really is whether we can accurately differentiate this group from the "generalized" group.
Also in practice, all we have often is that we can say for sure which type of seizures a patient shows, but not which type of epilepsy, as the latter is much more tricky to be unambiguously determined.
I think the authors are simply better off in talking about seizures only, and not about epilepsy types.
I would like to stress that the authors are technically so highly skilled that they might not wish to close their eyes from the fact that the mathematical formulae behind measures of effective connectivity are simply modelling statistical effects. They cannot demonstrate that there are indeed directional effects e.g. of information flow from one region to the other. With these analyses, we do not even know whether there is a physical connection between the two regions under question.
I know the connectivity hype has lead to many people believing that these statistical measures can show us anything about causal and directional relationships, but in fact this is just a statistic, and it might be true or wrong, as it relies on the data it gets. For the data the causal property is right, but not necessarily for the brain.
It is a nice idea to include the MAD for the information measures. I would assume this could have been included nicely in a figure but I know there is a limitation in the number of figures and tables you can include so no worries if that is not possible. I think in line 375 the value for pre is missing.
It is interesting to see the large variability in post, and that matches very much the reality: It is rather difficult to say when the seizure is over, and usually clinicians have a greater interest in determining the onset exactly than the end. Also, the end can vary a lot from seizure to seizure, that is indeed an interesting aspect.
I cannot find the statement about uncorrected p-values in lines 398-401? But I guess it would be best to mention this in/after lines 369-372, as well as in the discussion as a limitation.
Author Response
Please see the attachment.

This manuscript is a resubmission of an earlier submission. The following is a list of the peer review reports and author responses from that submission.
Round 1
Reviewer 1 Report
The authors present a framework for EEG analyses that addresses difficulties in attempting to characterize information transfer during peri-ictal phases of epileptic seizures. Overall, the ideas are interesting and the manuscript is well - written. However, there are some issues that need to be addressed
- The main goal was to identify a tool that is more robust than those currently in use. Although the theoretical basis for their method appears to be strong, the empirical evidence is less persuasive. In neither the generalized nor focal seizure groups is the difference between methods particularly striking. I agree with the assertion that volume conduction probably has little influence in generalized epilepsy - but that suggests a simpler framework is fine. I feel that the value of the proposed approach from an empirical standpoint would be beneficial
- It is important that the authors show example EEG traces, particularly for the focal seizures. It can be very difficult to identify EEG onset of a seizure. This could introduce variance into the dataset and thus influence the results.
- The data in the focal epilepsy group, using the new framework, seems to have an interesting bi-modality in Sj to it. There seems to be a subset in the post that is clearly different to the pre, and a subset with no differences. It would be interesting to try and identify features in the abnormal subset. This could help to define the types of clinical populations that the new framework would be relevant for.
- The age range of the patients is wide and there may be some developmental characteristics that deserve evaluation i.e are information parameters in a 3 year old brain likely to be consistent with those in an adult brain. This may deserve discussion but I suspect that there are insufficient patient numbers to formally evaluate
Reviewer 2 Report
In the manuscript, the authors have proposed a framework for the information-theoretic analysis of brain functional connectivity. The proposed framework demonstrated its practicality by evaluating for children suffering from focal and generalized epilepsy. According to the results of this manuscript, the proposed framework had a overall lower p-value compared to the scalp level and showed its superiority.
Here, I think this manuscript need to be revised on some information in terms of methods and results. It is listed in detail below.
- Major comments:
- About the reference paper [24], the authors called ‘CSPVARICA’, but is there a reason why this manuscript called ‘CSPMVARICA’ (line 76)?
- Please differentiate your paper from others. What is novelty in this paper?
- In Figure 1, can you add the description of the terms such as Q and N, and so forth.
- In Figure 3, 4, 5 and 6, what is the name of y-axis?
- Even if there is no moving from base to pre (line 320), why are format of Figure 3 and 5 different? I recommend modifying to the same format to help readers understand.
- As mentioned in line 403, this study consists of a relatively small number of subjects compared to other epilepsy studies. Can you add results using other public epilepsy dataset including subjects with various conditions?
- The subjects of this study is composed of young people under the age of 25, is there other research case like this? I wonder if this study can be applied to epilepsy cases of all ages.
- In Section 3, I think it would be better to emphasize the superiority of the proposed framework by using different evaluation metrics as well as p-value.
- Also, you need to compare performance with other frameworks, such as MVARICA, for the proposed framework.
- Minor comments:
- In line 117, please add ‘Eq.’ before the number in the formula. (e.g., ‘Eq.(2)’)
- As the above comments, please correct all the parts mentioned in the formula. For example, line 125, 129 and 132, and so forth.
- Although full name is shown in Section 1, please revise the acronyms such as ‘2.1. CSP’ and ‘2.2.3. ICA’ in the titles of each section to full names.
To conclude, I recommend “Reconsider after major revision”. I suggest that authors need to revise the raised issues.
